# Advancements in Hematopoietic Stem Cell Gene Therapy: A Journey of Progress for Viral Transduction

**DOI:** 10.3390/cells13121039

**Published:** 2024-06-15

**Authors:** Aurora Giommetti, Eleni Papanikolaou

**Affiliations:** 1Miltenyi Biotec B.V. & Co. KG, 51429 Bergisch Gladbach, Germany; aurorag@miltenyi.com; 2Faculty of Biology, University of Freiburg, 79104 Freiburg, Germany; 3Laboratory of Biology, School of Medicine, National and Kapodistrian University of Athens, 115 27 Athens, Greece

**Keywords:** gene therapy, hematopoietic stem cells (HSC), transduction, viral vectors, transduction enhancers, rare diseases

## Abstract

Hematopoietic stem cell (HSC) transduction has undergone remarkable advancements in recent years, revolutionizing the landscape of gene therapy specifically for inherited hematologic disorders. The evolution of viral vector-based transduction technologies, including retroviral and lentiviral vectors, has significantly enhanced the efficiency and specificity of gene delivery to HSCs. Additionally, the emergence of small molecules acting as transduction enhancers has addressed critical barriers in HSC transduction, unlocking new possibilities for therapeutic intervention. Furthermore, the advent of gene editing technologies, notably CRISPR-Cas9, has empowered precise genome modification in HSCs, paving the way for targeted gene correction. These striking progresses have led to the clinical approval of medicinal products based on engineered HSCs with impressive therapeutic benefits for patients. This review provides a comprehensive overview of the collective progress in HSC transduction via viral vectors for gene therapy with a specific focus on transduction enhancers, highlighting the latest key developments, challenges, and future directions towards personalized and curative treatments.

## 1. Introduction

In the area of medical science, the pursuit of innovative therapies has propelled researchers toward groundbreaking solutions to efficacious treatments that improve the patients’ quality of life. One such frontier is the transformative field of hematopoietic stem cell (HSC) gene therapy. HSCs are defined by the remarkable ability of long-term self-renewal and differentiation into multiple blood cell lineages, which elucidates their profound clinical significance. Over the years, HSC transplantation (HSCT) has become a well-established and widely utilized procedure for the treatment of congenital metabolic diseases and blood-related disorders. The first successful applications of allogeneic HSCT were achieved in the treatment of X-linked severe combined immunodeficiency (X-SCID) [1] and Wiskott–Aldrich syndrome (WAS) [2], where patients received stem cells from a compatible donor. Discovering its curative potential, notable advancements have been made in allogeneic HSCT that include the identification of donors with compatible human leukocyte antigen (HLA), the expansion of donor registries, and the possibility of alternative donor sources, such as haplo-identical donors who are half-matched to the recipient. The refinements in conditioning regimens have also contributed to improved patient outcome, as well as a more effective control of graft-versus-host disease (GvHD) that involves the selective depletion of α/β T cells and naïve T cells [3]. Nevertheless, the successful applications of allogeneic HSCT can be constrained by the availability of suitable donors and a potential risk of morbidity and mortality due to the use of HLA-mismatched individuals. Autologous HSCT represents a significant leap toward overcoming the risk of graft rejection and additional complications arising from alloreactivity [4]. Therefore, autologous HSCT, in combination with HSC gene therapy, has been explored as an alternative therapeutic strategy to treat not only hematological malignancies but also inherited diseases, including severe immunodeficiencies, hemoglobinopathies, and metabolic disorders. In this approach, the patient’s own stem cells are harvested, cultured ex vivo, and genetically modified before being reinfused into the patient following an appropriate conditioning regimen to deplete progenitor and differentiated cells in the bone marrow and to favor engraftment. In this context, ex vivo HSC genetic engineering can be performed either through transduction with viral vectors delivering the therapeutic gene of interest or by targeted genome editing approaches that allow site-specific genome modifications depending on the disease. This review will analyze the current state of hematopoietic stem cell gene therapy, addressing the latest advancements, challenges, and therapeutic potentials of this application. After an overview of the gene transfer processes, we focus on the approaches that generated the first marketing authorizations, i.e., the vector-based modifications of HSCs and their optimization regarding efficacy and safety aspects, concluding with an overall discussion about the prospective developments in HSC gene therapy.

## 2. Gene Delivery in Hematopoietic Stem Cells

### 2.1. Overview

The most extensively investigated gene transfer techniques in hematopoietic stem cells to date are based on the ex vivo approach. This type of gene therapy involves the collection of HSCs from the patient that undergo genetic manipulation through viral transduction or gene editing to restore the correct phenotype. Following a preconditioning regimen, the engineered cells are infused back into the patient’s body, where they self-renew and differentiate generating a long-term reservoir of modified HSCs giving rise to multiple blood lineages. This strategy allows the performance of the manipulation process in a controlled environment that enables the monitoring of cell characterization and functionality before transplantation, and it potentially represents a one-time curative treatment due to the engraftment capacity of gene-corrected HSCs. Therefore, it is a promising therapy to tackle hematological disorders and immune aberrations, as confirmed by the marketing authorization for advanced therapy medicinal products (ATMPs) based on engineered HSCs, namely Libmeldy™ for pediatric metachromatic leukodystrophy (MLD), Strimvelis^®^ for the treatment of severe combined immunodeficiency due to adenosine deaminase deficiency (ADA-SCID), Zynteglo™ for β-thalassemia, Lyfgenia™ for the treatment of sickle cell disease (SCD), and Skysona^®^ for early cerebral adrenoleukodystrophy (CALD) (Table 1).

### 2.2. Gammaretroviruses

Ex vivo gene manipulation employing viral vectors as delivery tools has been a major player in the field of HSC gene therapy. Following HSC collection and enrichment of CD34^+^ cells, they undergo genetic modification through viral vector transduction before being reinfused to the host, where their engraftment will result in a sustained transgene function. This method exploits the innate ability of viruses to efficiently internalize their own genome into the target cells; however, the viral vectors are engineered from wildtype viruses by removal of most of the genes encoding for viral proteins from the viral genome to make them replication incompetent. A variety of viral vector classes have been used in several clinical trials conducted so far, including members of the *Retroviridiae* family (gammaretroviruses and lentiviruses), adenoviruses, and adeno-associated viruses, and each of these platforms has highlighted advantages and complications in preclinical and clinical testing phases. Gammaretroviruses (γRV) were initially the first vectors assessed for gene therapy and their popularity did not decrease overtime, mainly due to their low immunogenicity, ability to integrate the viral genome into the host cells, and the high efficiency of transduction in actively dividing target cells. Therefore, considering the quiescent state of HSCs, they are pre-stimulated to induce the cell cycle for an effective γRV transduction. However, gammaretroviral vectors preferentially integrate near transcriptional start sites and within CpG islands, and they have affinities toward proto-oncogenes, potentially leading to insertional oncogenesis and serious adverse events such as malignant transformation, as this was demonstrated in the original X-SCID study [5]. Thus, the constrained efficacy of γRV gene transfer into HSCs and the risk of genotoxicity prompted the advancement of self-inactivating lentiviral vectors (LVs) as a preferred delivery system due to their improved safety profile. 

### 2.3. Lentiviruses

Lentiviruses employed in the clinic are usually based on human immunodeficiency virus (HIV) 1 devoid of its structural genes and commonly pseudotyped with a different envelope, and they have the ability to infect non-replicating cells due to the transfer of viral vector DNA in the nucleus via nuclear pores, allowing a faster transduction process and also harnessing their larger packaging capacity of up to 9 kb. Although the integration sites of LVs are generally unpredictable with hotspots in active transcription units, Biffi et al. [6] suggested that they cluster in the megabase-wide genomic regions without accumulation in specific genomic regions, in contrast to genotoxic integration sites that are not distributed along chromosomes but come across as isolated sharp peaks and always target a single gene, which is the culprit of oncogenesis. This study [6] provided a more comprehensive understanding of the preferential or common integration sites among retroviral vectors and showed that LVs present markedly consistent integration pattern. Alternative vector systems, such as adenoviruses and adeno-associated viruses, have encountered restricted success when applied to HSCs thus far due to the strong innate and adaptive immune responses induced by viral infection [7] and due to the challenging transient transgene expression even with high multiplicity of infection [8], respectively. Overall, genetic engineering of HSCs via viral vector transduction has been shown to be a crucial tool in the clinical setting for HSC gene therapy, achieving curative treatments in several clinical trials [9,10,11] and successful marketing approvals [12,13,14]. 

### 2.4. Genome Editing

Nonetheless, alongside HSC transduction, the identification and adaptation of programmable molecules such as nucleases, base editors, and prime editors have enabled targeted genetic modifications, allowing a major step forward in the field of genome editing. In this context, site-specific endonucleases are used to introduce a double-stranded break at a desired location at the DNA level that will recruit DNA repair proteins to correct the damage, establishing specific genetic changes which, depending on the type of indication, could result in gene disruption, gene correction or gene insertion. Although a detailed analysis of these methods falls beyond the scope of this review, it is worth mentioning the recent marketing approval of Casgevy™, the first CRISPR-Cas9 gene editing therapy, which aims to cure SCD and transfusion-dependent β-thalassemia (TDT) [14]. However, it should be noted that there is a growing body of literature evaluating the unexpected adverse events of the use of CRISPR technology for HSC modifications, including on-target genotoxicity such as deletions, translocations, micronuclei formation [15], and decreased long-term in vivo engraftment in terms of clonal dynamics [16]. A more detailed description of the CRISPR complications is presented in the paragraph “Safety Considerations” (see below).

## 3. Advancements in Transduction Technologies 

### 3.1. Viral Vector Engineering 

Even though HSC gene therapy utilizing viral vectors has evolved as a therapeutic alternative for various inherited diseases and gained success over the decades, achieving stable and clinically relevant gene transduction in hematopoietic stem cells still presents a significant challenge. Successful viral transduction of HSCs is hampered by several barriers, including low expression levels of viral receptors on the HSC surface, inefficient viral entry, as well as the presence of cellular restriction factors that inhibit viral replication. Due to their quiescent nature, differently from other target cell types such as T cells or natural killer cells, an efficient uptake of vectors by HSCs requires high multiplicity of infection (MOI) resulting in increased viral vector doses as the cell number grows, which can increase the risk of adverse events, and it is particularly relevant for large-scale production of engineered HSCs that will be translated into clinical applications.

A promising solution that is gaining steam is to enrich an HSC subpopulation by sorting the CD34^+^CD38^−^ cells which represent the small proportion of HSCs that actually contribute to long-term hematopoiesis, allowing a reduction in the amount of viral vector required for transduction without impacting the vector copy number (VCN) and the unification of the final characteristics of the infused HSCs that come from different sources such as bone marrow or mobilized peripheral blood. However, although there was initial enthusiasm, this approach was not widely applied in the clinic due to the difficulties in enriching pure CD34^+^CD38^−^ populations, which made the overall process laborious, lengthy, and eventually inefficient. 

In a different approach, the quest for more effective gene transfer techniques employed evaluation of several vector designs with a particular focus on the investigation of different envelope proteins for vector pseudotyping to check which one can increase their binding and uptake by target HSCs. Initial attempts to pseudotype the vectors with retroviral envelopes deriving from amphotropic murine retrovirus or the Gibbon Ape Leukemia Virus (GALV) led to low viral titers which triggered their replacement with more efficient envelopes, most notably the glycoprotein derived from the vesicular stomatitis virus (VSV-G) that confers broad tropism and enhances the vector stability, enabling effective HSC transduction. Moreover, VSV-G pseudotyping not only allows the concentration of vectors at high titers, but its robust fusion activity also facilitates entry into HSCs, exploiting the abundantly expressed low-density lipoprotein receptor (LDL-R) [17,18], bypassing the need for specific uncommon cellular receptors, which may be a limiting factor for successful transduction. Furthermore, a novel envelope protein derived from baboon endogenous retrovirus (BaEV) holds significant promise for HSC gene therapy applications. The BaEV envelope offers several benefits, including its unique natural tropism for human CD34^+^ hematopoietic stem cells and a favorable safety profile [19]. This inherent specificity reduces the risk of off-target effects and enhances the efficiency of gene delivery, making it particularly well suited for HSC transduction. In addition, although the titer measurements were lower than the VSV-G pseudotyped viral vectors, BaEV is more effective at an equivalent amount [19] and displays minimal immunogenicity, reducing the likelihood of immune responses that could compromise the success of the therapy. 

On the basis of the promising envelope improvement, researchers are poised to develop safer and more targeted approaches to manipulate vectors for HSC transduction. A cutting-edge strategy to improve the vector design consists of the integration of recombinant membrane proteins leading to the expression of cytokines (e.g., stem cell factor and thrombopoietin) on the surface of the vector virions that will specifically recognize and bind to the respective receptors present on the HSC surface (c-kit, c-mpl). This system was designed to promote high levels of transduction of the most immature CD34^+^ cells, crucial for clinical application, with a selective and minimal HSC stimulation that is already supplied by the cytokines expressed on the virions, avoiding the addition of hematopoietic growth factors in the medium [20]. In agreement, a greater preference of lentiviruses in clinical trials has been currently observed, compared to their retroviral or adenoviral counterparts due to the ability of LVs to integrate into non-dividing cells that do not require prolonged cytokine stimulation to activate the HSC cell cycle before transduction, thereby circumventing intense cell proliferation in culture that could progressively compromise engraftment potential [21]. The current focus mainly lies in achieving an efficient lentiviral vector transduction of long-term repopulating quiescent HSCs, which are resistant to genetic manipulation but an ideal gene therapy target, with minimal in vitro culturing to avoid extensive cell stimulation and cell cycle commitment [22].

### 3.2. Transduction Enhancers 

Along with the progress in engineered vector design and ex vivo culture conditions, several reagents have been tested aiming to achieve a clinically translatable transduction efficiency without interfering with HSC self-renewal and differentiation. A promising strategy is the addition of transduction enhancers which encompass a diverse array of small molecules that modulate cellular pathways involved in viral entry, intracellular trafficking, endosomal escape, or nuclear import (Figure 1). An overview of several transduction enhancers is shown in Table 2.

Prostaglandin E2 (PGE2) has been investigated as an adjuvant that promotes lentiviral transduction of CD34^+^ cells and modestly increases the VCN both in vitro and in an NOD/SCID xenotransplantation mouse model without evidence of in vivo toxicity [23] while reducing the duration of ex vivo culture. The mechanism leading to higher transduction levels is still under investigation, but PGE2 may act by improving the reverse transcription and hence replication of the vector inside the cell prior to nuclear entry and integration, since an increase in late RT copies was detected within 6 h after transduction [24,25]. Additional studies reported the beneficial effect of PGE2 on HSC transduction and VCN [26,27]; however, Poletti et al. [28] showed that it causes a significant reduction in stem cell clonogenic capacity once transplanted in humanized immunodeficient mice in a competitive repopulation assay despite its safety in increasing cord blood engraftment being demonstrated in the clinic [29] and its favorable effects being corroborated in a clinical trial for the treatment of Hurler disease [30]. It should be noted, however, that there was a lot of skepticism in the gene therapy field about the use of PGE2 because of the preexisting evidence of potential reduction in stemness [28]. The leukemic events in the bluebird clinical trial for SCD [31] also corroborated the initial doubts on the grounds of generation of leukemic phenotypes as a result of graft failure since the percentage of leukemic events in the gene therapy setting was equivalent to the leukemias presented in the allogeneic setting for SCD when graft failure is observed [32]. In particular, the working hypothesis was that after rejection or gene therapy, the stress from switching from homeostatic to regenerative hematopoiesis by autologous cells drives clonal expansion and leukemogenic transformation of preexisting premalignant clones, eventually resulting in hematological dysplasias. Nevertheless, in December 2023, the U.S. Food and Drug Administration (FDA) granted marketing authorization for Lyfgenia™, the lentiviral approach to treat SCD by bluebird bio, but with a warning for blood cancer [33]. It should be also noted that, in view of the exact same manufacturing process utilized both in the case of Lyfgenia™ and Zynteglo™, more gravity is given to the specific pathophysiology of SCD and not to the vector–cell interactions. Finally, the efficiency of lentiviral vector transduction on HSCs exhibited a significant increase also when PGE2 was tested in combination with polybrene [26], a surfactant polycation widely and successfully utilized as a transduction enhancer from the early days due to its interaction with the negatively charged cellular membrane which leads to charge shielding between the vector and the cell surface [34], but this approach did not reach clinical applicability. 

Traditionally, protamine sulfate represents another cationic additive which produces an optimal transduction-enhancing effect by neutralization of the cell membrane charge, and its approval for human use by the FDA, together with its low toxicity on a range of cell types [35], highlights its versatility and potential for clinical translation. A similar mechanism of action is observed when poloxamer is supplemented in culture, where it influences the physiochemical properties of the cell membrane, also promoting transmembrane transport. Different sizes of poloxamers have been investigated, among which P118, P338 [36], and P407 which result in a similar increment in both the percentage of transduced cells and the number of vector copies per cell without significant toxicity [25]. Lately, the most commonly used transduction enhancer is termed LentiBOOST™ and consists of a combination of poloxamer 338 and Pluronic F108 and is considered an entry enhancer because it seems to increase the permeability of the target cell surface facilitating the entry of viral particles [37]. LentiBOOST™ outperformed the aforementioned enhancers, leading to a strong effect in terms of vector expression at low MOI with an acceptable VCN increase [38] and maintaining HSC differentiation potential, and was also demonstrated in xenotransplantion experiments [27] with no signs of toxicity in vivo [39]. Another peptide that enables high levels of gene transfer with various retroviral and lentiviral pseudotypes into CD34^+^ HSCs is Vectofusin-1^®^ [40]. Through the formation of alpha-helical nanofibrils, it fosters the adhesion of viral vectors to targeted receptors on the cell surface and facilitates endocytosis, ultimately leading to increased transduction efficiency of HSCs in vitro that is preserved also in their progeny (T and B cells) after engraftment in an NSG mouse model [41]. Specifically, Vectofusin-1^®^ does not alter the cell viability and functionality and the safety of the transduction process, and since it is soluble in water, it allows avoidance of the pre-coating step required for similar compounds such as Retronectin, making it an ideal candidate for clinical settings and scalable gene therapy protocols [42]. Furthermore, a deeper understanding of the mechanisms regulating HSC proliferation, self-renewal, and quiescence has led to the detection of rapamycin, a macrolide compound with immunosuppressive properties, as a potential transduction enhancer for HSC engineering. While the precise underlying mechanisms still need to be elucidated, rapamycin promotes efficient viral transduction of both human and murine HSCs via the inhibition of the mTOR signaling pathway, significantly boosting the frequency of long-term engrafting cells in mice [43] and ex vivo long-term hematopoietic reconstitution [44]. Moreover, rapamycin’s well-established safety profile and clinical use in other therapeutic contexts, including prevention of allograft rejection and cancer treatment, underscores its potential application in clinically relevant viral transduction protocols. 

**Table 2 cells-13-01039-t002:** List of reagents employed to enhance transduction efficiency.

Reagent	Mechanism of Action	Side Effects	Side Effects in Gene Therapy	Clinical Applications in Gene Therapy	References
Prostaglandin E2	Improvement of reverse transcription (under investigation)	Nausea, vomiting, diarrhea, abdominal pain	Reduction of HSC clonogenic potential	Hurler syndrome, β-thalassemia	[21,26,28]
Protamine sulfate	Lower charge repulsion between the vector and the cell surface	Low blood pressure, allergic reactions, vomiting	Cell toxicity (concentrations higher than 10 µg/mL)	N/A	[33]
Poloxamers	Membrane fluidization and reduction in electrostatic barriers	Dehydration, abdominal discomfort	N/A	N/A	[34]
LentiBOOST™	Increased permeability of the target cell surface	N/A	N/A	X-SCID, Artemis-SCID	[35]
Vectofusin-1^®^	Enhanced adhesion and fusion of viral particles to the cell membrane	N/A	N/A	N/A	[38]
Rapamycin	Inhibition of mTOR signaling pathway (immunosuppression)	Anemia, increased blood pressure, muscle pain	N/A	N/A	[41,43]
Cyclosporin A Cyclosporin H	Inhibition of cyclophilin A (immunosuppression) Inhibition of IFITM3	Blurred vision, back pain, dizziness, decreased appetite	N/A	N/A	[43,45,46]

N/A: not applicable.

Along with rapamycin, cyclosporin A (CsA) is an additional immunosuppressive compound that acts by inhibiting the activity of the cellular protein cyclophilin A, which is known to interact with the viral capsid protein of retroviruses and lentiviruses during transduction. By blocking this interaction, CsA enhances the efficiency of viral vector entry into target cells, including HSCs, thereby improving transduction efficiency without adversely affecting their colony-forming capacity. Importantly, increased transduction efficiencies were maintained long term in vivo and no negative effects on HSC engraftment were observed [45]. Additionally, CsA has been shown to mitigate the inhibitory effects of cellular antiretroviral restriction factors, such as TRIM5α, on viral transduction, further enhancing gene delivery to HSCs. These results were corroborated by Evans and colleagues [46], who reported that TRIM5α transcript levels in human CD34^+^ cells correlate with donor variability in transduction efficiency with lentiviral vectors. From the same class of compounds of CsA, cyclosporin H (CsH) has gained attention as a transduction enhancer operating with a mechanism of action similar to CsA. In detail, CsH reduced the innate resistance mechanism against LV infection performed through the interferon-induced transmembrane protein 3 (IFITM3) constitutive inhibition of viral entry by degrading IFITM3, leading to significant improvement in gene transfer levels both in murine and human HSCs [47,48]. 

Overall, the use of transduction enhancers will have profound implications for clinical practice. By enhancing the efficiency of gene delivery, these strategies allow minimization of the multiplicity of infection of the viral vector, improving safety and therapeutic efficacy of HSC gene therapy applications. Furthermore, the development of targeted and customizable transduction enhancers may enable tailored approaches for specific patient populations and disease contexts, minimizing off-target effects and optimizing treatment outcomes. 

## 4. Safety Considerations 

Ensuring the safety of HSC transduction is paramount to sustained clinical efficacy and the long-term success of therapeutic interventions and it represents a crucial step not only for manufacturing of engineered HSCs but also from a pure clinical perspective. Indeed, the HSC gene therapy field was initially severely hampered because of safety concerns deriving from insertional mutagenesis (creating the risk of leukemia) and/or immunogenicity. 

### 4.1. Genotoxicity and Leukemias 

In the notorious French X-SCID clinical trial, four out of seven patients treated initially with gamma RV developed lymphocytic leukemia [49,50], which was associated with vector integration in the vicinity of the LMO2 gene, leading to its upregulation, which was suggested to be the determining event for the onset of the blood cancer. The development of malignancies resulted in a temporary interruption of this trial, which eventually resumed, but only for patients who had failed standard transplant therapy. In the early 2000s, other gene therapy trials were also delayed due to the potential risk of leukemia but eventually continued as the risk-versus-benefit ratio was deemed to be in favor of the patients [51].

Unfortunately, leukemias were not observed only during the French X-SCID trial. Cancer transformation was observed in other clinical trials employing γRV vectors for X-linked Chronic Granulomatous Disease (X-CGD) [52] and WAS [53] as a result of vector integration close to and activation of proto-oncogenes. At that time, the field’s response focused on the following actions: a) further advancement of viral vector engineering within the context of self-inactivating (SIN) LVs as a vehicle for gene delivery and b) a deeper understanding of the vectors’ integration sites, alongside rigorous preclinical safety testing to predict potential adverse effects and mitigate the long-term risk of genotoxicity. Additionally, in vitro assays such as the In Vitro Immortalization assay (IVIM) and the Surrogate Assay for Genotoxicity Assessment (SAGA) have been developed in an effort to predict or quantify the pre-clinical genotoxicity of integrating vectors. Notably, IVIM quantifies the mutagenic potential of retroviruses based on the acquisition of a proliferation advantage under limiting dilution conditions of murine hematopoietic stem and progenitor cells transduced with mutagenic vectors [54]. Although this approach is relatively specific for the detection of mutants with insertions near the *Mecom* locus (also known as *Evi1*) or its close relative *Prdm16*, both of which were shown to be clinically relevant as inducers of clonal imbalance in clinical trials for X-CGD [55,56], X-SCID [49], WAS [57], it has been accepted by regulatory authorities as part of the pre-clinical safety assessment. A more accurate prediction is performed with the SAGA approach that classifies integrating retroviral vectors using machine learning algorithms to detect the activation of gene expression pathways connected to oncogenesis during the course of in vitro cell immortalization [58]. However, due to the specific culture conditions, both assays present an intrinsic myeloid bias, and thus Bastone and colleagues [54] have introduced the SAGA-XL assay that follows a similar bioinformatic strategy enabling the identification of lymphoid genotoxicity predictors. Notably, it should be mentioned that the onset of leukemias in HSC gene therapy is not always the result of vector-induced insertional mutagenesis, since in the case of the Lyfgenia™ trials, the leukemic blasts were devoid of vector genetic material, clearly suggesting that the dysplasias were independent of the vector in this specific setting and were rather associated with the overall stemness/fitness of the graft and/or the specific pathophysiology of SCD. 

Genotoxicity poses concerns also in the CRISPR field, as a growing line of evidence shows the occurrence of unexpected on-target genotoxic events, including deletions and chromosomal translocations, which may compromise the genomic integrity and functionality of edited HSCs. Studies, such as Kosicki et al. [59], have highlighted that DNA breaks introduced by CRISPR-Cas9 editing can resolve into onsite large deletions as well as crossover events and lesions distal to the cut site, which may constitute a first carcinogenic ‘hit’ in stem cells and progenitors that have a long replicative lifespan [59]. Further works reported that p53-mediated DNA damage response activated by double-strand breaks induced by CRISPR could lead to selection for cells with mutations in the p53 pathway [60], potentially contributing to oncogenesis. Recently, Lee et al. [16] have shown that the homology-directed repair (HDR) pathway induced by CRISPR-Cas9 led to decreased short- and long-term multilineage HSC engraftment and graft clonality in a competitive rhesus macaque autologous transplantation model. In the same study, the authors demonstrated that CRISPR/HDR-edited cells showed lower viability, cell proliferation, and markedly decreased long-term engraftment compared to lentiviral transduced cells, suggesting increased toxicity of editing [16]. It should be noted, however, that the latest results by Zeng and colleagues [15] indicated a better outcome for the CRISPR-engineered HSCs in terms of genotoxicity and micronucleation in the context of short-term ex vivo culture in the absence of cytokines (hence avoiding the induction of cell cycle through ex vivo cytokine stimulation). 

### 4.2. Immunogenicity 

Other possible obstacles that could trigger undesirable events can arise from innate and adaptive immunity against reagents used during manufacturing and immune reactions against neoantigens introduced into HSCs by genetic engineering [61] due to gene disruptions and/or translocations. To address these concerns, which are triggered by the transduction/gene editing per se and are expressed long term after the administration of the genetically corrected graft, will require further in-depth investigations, always taking into consideration the vector system, the engineering approach, and the transplantation settings applied in each different scenario.

## 5. Conclusions and Future Perspectives

Over the last twenty years, the HSC gene therapy field has witnessed notable clinical achievements that resulted in remarkable marketing approval of ATMPs by the EMA and/or FDA. A scientific breakthrough came in 2016 when Strimvelis^®^ received marketing authorization for the treatment of ADA-SCID, employing a gamma retroviral vector carrying the sequence of the therapeutic gene. The approval of Zynteglo™ offers patients with TDT the possibility to be treated with a lentiviral vector encoding a β-globin transgene, which is mutated or absent in these patients. Lastly, there was also the marketing authorization of Skysona^®^ for early CALD. In terms of the availability and prices of these drugs, it is necessary to underline that ex vivo HSC gene therapy is an extremely personalized treatment that presents significant technical challenges. A prime example is Strimvelis^®^, which, owing to its fresh formulation, can be administered only at the approved manufacturing facility in Europe in Milan, where the patient and their family have to stay for around 4 months. In this specific case, the overall costs are covered by the national insurance of the patient’s country, but this does not apply for Zynteglo™ and Skysona^®^, which, despite obtaining marketing authorization, have been withdrawn from Europe due to their extremely high price tag per patient [62]. Besides gene therapy products, another milestone treatment for TDT and for severe SCD patients is Casgevy™, the first gene editing technology based on CRISPR-Cas9 system on the market, although discussion around the reimbursement from public health budgets or insurance companies is still ongoing. Thus, although there are significant number of marketing authorizations in spite of the aforementioned limitations, several HSC gene therapy trials involving viral vectors and genome editing are still ongoing (Table 3). These ongoing trials are diversified compared to the initial ones, either by differential patient stratification (e.g., in the KL003 trial, TDT patients are stratified based on the levels of serum ferritin) or by addition of transduction enhancers (NCT03538899, NCT01306019), indicating that while significant progress has been made in improving HSC transduction efficiency and long-term safety, challenges remain, limiting the clinical applicability of gene therapy. To this end, one might argue that the development of transduction enhancers holds promise for overcoming barriers to HSC transduction towards improving therapeutic outcomes because of the long-standing clinical experience with lentiviral vectors in the field.

Finally, advancements in gene editing technologies with designer nucleases offer exciting opportunities for precise genome modification in HSCs, paving the way for personalized and curative treatments. In this context, over the last decade, nanoparticles (NPs) have also emerged as an attractive tool to deliver therapeutic agents with sharp specificity and versatility. In particular, NPs could be equipped with targeting motifs specific to HSCs and can potentially overcome the need for ex vivo manipulation of patient HSCs [63]. Among the investigated NPs, lipid nanoparticles (LNPs) have been reported to substantially decrease electroporation-induced toxicity and to generate higher yields of edited cells [64]. 

By leveraging emerging technologies and prioritizing early development of options that will feasibly become drugs which are economically competitive compared to the standard of care treatment, further products will reach the market to bridge the gap between bench research and patient’s bedside.

To conclude, toward safer gene therapies for blood disorders, the focus should be on the potency of the graft both in terms of engraftment and clonogenic capacity and in terms of functionality, i.e., its ability to produce a functional amount of the therapeutic protein.

As for any other cell product manufactured by genetic modification, the HSC-engineered grafts have to comply with the overall EMA/FDA regulation for ATMPs, which pose a long-term follow-up for at least 15 years to monitor the persistence of their clonal dynamics to detect potential adverse events (e.g., leukemogenesis or blood dysplasias) as a result of clonal hematopoiesis due to the engineering approach (vector transduction or genome editing). Therefore, risk mitigation performed both by applying safety strategies during manufacturing (SIN vectors, optimized ex vivo protocols, use of transduction enhancers, etc.) and by implementing careful monitoring and long-term pharmacovigilance measures has the potential to promote gene therapy approaches to medical routine. 

Finally, further attempts should aim toward alleviating the “financial” toxicities which currently severely limit the wider applicability of these therapies. 

## Figures and Tables

**Figure 1 cells-13-01039-f001:**
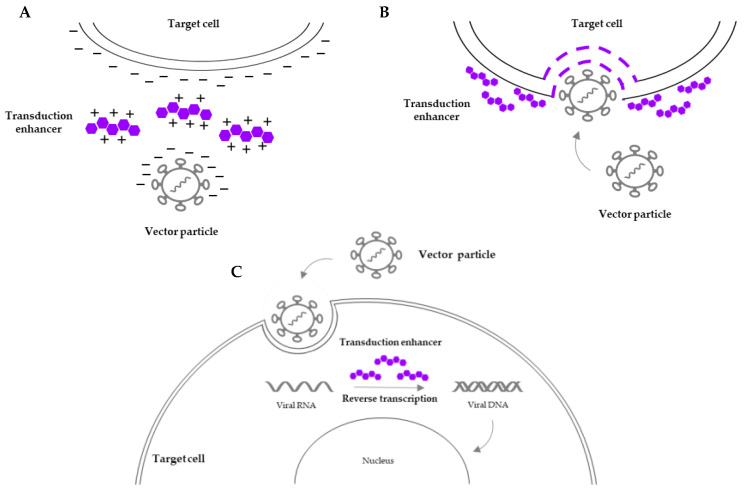
Overview of the mechanisms of action of transduction enhancers. (**A**) Lower charge repulsion between the vector particle and the target cell surface facilitates cell-to-vector interaction. (**B**) Increased permeability of the target cell surface membrane facilitates viral entry. (**C**) Influence on intracellular processes (e.g., reverse transcription of viral RNA) prior to vector integration into the host genome facilitates vector trafficking and integration.

**Table 1 cells-13-01039-t001:** Hematopoietic stem cell gene therapy products approved on the market.

Product Name: Generic (Trade)	Applications	Manufacturer	Mechanism of Action	Approval Agency (Year)
STRIMVELIS^®^	ADA-SCID	Orchard Therapeutics	ADA gene addition via gamma retrovirus	EMA (2016)
Betibeglogene autotemcel (ZYNTEGLO™)	Transfusion-dependent Β-thalassemia (TDT)	bluebird bio, Inc.	βA-T87Q-globin gene addition via lentivirus	EMA (2019) * FDA (2022)
Atidarsagene autotemcel (LIBMELDY^®^)	Metachromatic leukodystrophy (MLD)	Orchard Therapeutics	ARSA gene addition via lentivirus	EMA (2020) FDA (2024)
Lovotibeglogene autotemcel (LYFGENIA™)	Sickle cell disease (SCD)	bluebird bio, Inc.	βA-T87Q-globin gene addition via lentivirus	FDA (2023)
Exagamglogene autotemcel (CASGEVY™)	TDT, SCD	Vertex Pharmaceuticals CRISPR Therapeutics	CRISPR/Cas9 technology	EMA (2023) FDA (2024)
Elivaldogene autotemcel (SKYSONA^®^)	CALD	bluebird bio, Inc.	ABCD1 gene addition via lentivirus	EMA (2021) * FDA (2022)

* Withdrawn at the request of the marketing-authorization holder.

**Table 3 cells-13-01039-t003:** Currently ongoing hematopoietic stem cell gene therapy clinical trials.

Clinical Trial Registry Number	Disease	Intervention	Sponsor	Phase
NCT04797260	RAG1-SCID	Autologous CD34^+^ cells transduced with the pCCL.MND.coRAG1.wpre LV	Leiden University Medical Center	I/II
NCT05071222	Artemis-SCID	Autologous CD34^+^ cells transduced with the G2ARTE LV expressing the DCLRE1C cDNA	Assistance Publique—Hôpitaux de Paris/Genethon	I/II
NCT02559830	MLD, ALD	Autologous CD34^+^ cells transduced with a LV encoding the human ARSA(for MLD)/ABCD1(for ALD) cDNA	Shenzhen Second People’s Hospital	I/II
NCT05860595	TDT	Autologous CD34^+^ cells transduced with the βA-T87Q-globin gene LV (KL003)	Institute of Hematology and Blood Diseases Hospital, China/Kanglin Biotech	N/A
NCT05762510	TDT	Autologous CD34^+^ cells transduced with the GMCN-508B (LentiRed) LV	First Affiliated Hospital of Guangxi Medical University	Early I
NCT05432310	ADA-SCID	Autologous CD34^+^ cells transduced with the EFS-ADA LV encoding the ADA enzyme	University of California, Los Angeles	I/II
NCT06149403	Hurler syndrome	Autologous CD34^+^ cells transduced with LV encoding the human IDUA gene	Orchard Therapeutics	III
NCT05265767	Hemophilia A	Autologous CD34^+^ cells transduced with LV encoding a novel coagulation factor VIII transgene	Christian Medical College, Vellore, India	I
NCT03818763	Hemophilia A	Autologous CD34^+^ cells transduced with LV encoding the *ITGA2B* gene promoter for ectopic expression of human B-domain-deleted factor VIII	Medical College of Wisconsin	I
NCT06155500	SCD	Observational: long-term follow-up of patients treated with CRISPR/Cas9-edited HSPCs from NCT04443907	Novartis Pharmaceuticals	I
NCT01306019	X-SCID	Autologous CD34^+^ HSC with VSV-G pseudotyped LV CL20- 4i-EF1alpha-hgammac-OPT	National Institute of Allergy and Infectious Diseases (NIAID)	I/II
NCT03538899	Artemis-SCID	Autologous CD34^+^ cells transduced with LV (AProArt) encoding the corrected DCLRE1C gene	University of California, San Francisco	I/II
NCT05757245	TDT	Autologous CD34^+^ cells transduced with GMCN-508A LV	First Affiliated Hospital of Guangxi Medical University	I
2014-000274-20	WAS	Observational: long-term follow-up of patients treated with w1.6_hWASP_WPRE (VSVg) LV transduced autologous HSCs	Genethon	II
2019-004266-18	TDT	Observational: long-term follow-up of patients treated with βA-T87Q LV (LentiGlobin BB305) transduced autologous HSCs	bluebird bio, Inc.	III
2020-000517-33	Leukocyte adhesion deficiency I	Autologous CD34^+^ cells transduced with LV encoding the ITGB2 gene	Rocket Pharmaceuticals, Inc.	I/II
2017-001366-14	TDT	Observational: long-term follow-up of patients treated with GSK2696277	GlaxoSmithKline Research and Development	II
2017-002430-23	Hurler syndrome	Autologous CD34^+^ cells transduced with IDUA LV encoding the human α-L-iduronidase gene	Ospedale San Raffaele	I/II
2018-001404-11	Glioblastoma multiforme	Autologous CD34^+^ cells transduced with LV encoding the interferon-α2 gene	Genenta Science S.r.l	I/IIa
2013-002245-11	Hemoglobinopathies	Observational: long-term follow-up of patients treated with LentiGlobin BB305 Drug Product	bluebird bio, Inc.	III

Data taken from www.clinicaltrials.gov and www.clinicaltrialsregister.eu (both accessed on 31 May 2024) for recruiting and ongoing (respectively) clinical trials based on the search term ‘haematopoietic stem cell’ AND ‘gene therapy’. N/A: not applicable.

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
