# Peer review of "Advancements in Hematopoietic Stem Cell Gene Therapy: A Journey of Progress for Viral Transduction"

_cells, 2024, doi:10.3390/cells13121039_

Round 1

Reviewer 1 Report

Comments and Suggestions for Authors

In this review manuscript the authors briefly summarized the current status of the engineered HSC therapy. This manuscript highlights some critical points including gene delivery in hematopoietic stem cells, advancements in transduction technologies, safety considerations which provides the essential knowledge for the readers that work in this area.

Following are minor suggestions:

1.          Please use a table to summary/update the authorized HSC related cell and gene products and collect additional information about the products, e.g., the product were approved by EMA or FDA; approved date (year), and sponsor. It will let reader to quickly catch the key information.

2.          Again, in the section about the transduction enhancers, it would be great to have a table to list the reagents to enhance the transduction efficiency. For example,

Reagent

Mechanism (if known)

Side effect

Clinical application

Reference

Author Response

In this review manuscript the authors briefly summarized the current status of the engineered HSC therapy. This manuscript highlights some critical points including gene delivery in hematopoietic stem cells, advancements in transduction technologies, safety considerations which provides the essential knowledge for the readers that work in this area.

We thank the Reviewer for this comment.

Following are minor suggestions:

  1. Please use a table to summary/update the authorized HSC related cell and gene products and collect additional information about the products, e.g., the product were approved by EMA or FDA; approved date (year), and sponsor. It will let reader to quickly catch the key information.

We thank the Reviewer for this comment. As per the Reviewer’s suggestion, we have included Table 1 on pages 2 and 3.

  1. Again, in the section about the transduction enhancers, it would be great to have a table to list the reagents to enhance the transduction efficiency. For example,

Reagent

Mechanism (if known)

Side effect

Clinical application

Reference

We thank the Reviewer for this comment. As per the Reviewer’s suggestion, we have included Table 2 on page 7.

Reviewer 2 Report

Comments and Suggestions for Authors

Authors discuss a topic that is timely but also covered by many excellent recent reviews, so it is unclear what exactly this review will add to the literature. Nevertheless, there are some points that need to be addressed anyway:

1.       The authors are overly optimistic about gene editing using CRISPR. There is no  a growing literature about unexpected side effects of CRISPR in HSC, including on target side effects, such as deletions, translocations etc. and recent work in Cell stem Cell, by the Dunbar lab on engraftment of HSC.

2.       I am missing the work of the Hannover group on IVIM and recently SAGA assays as important safety assays for LV transduction in HSC

3.       There are many more indications for which HSC based gene therapy is in clinical trails than authors discus, notably Artemis-SCID, RAG1-SCID and Fanconi anemia. This should be discussed and relevant work cited. Perhaps adding a table of ongoing trails for HSC GT would be useful

Author Response

Authors discuss a topic that is timely but also covered by many excellent recent reviews, so it is unclear what exactly this review will add to the literature. Nevertheless, there are some points that need to be addressed anyway:

We thank the reviewer for this comment. To facilitate further understanding of the scope of this review we slightly amended the abstract by clarifying the review’s focus on transduction enhancers and we also added the word “latest” in the abstract. These changes can be now found on page 1, lines 20-22. Addressing the reviewer respectfully, we are not aware of a review with a specific focus on lentiviral transduction enhancers which prompted us to draft the current manuscript.

  1. The authors are overly optimistic about gene editing using CRISPR. There is no  a growing literature about unexpected side effects of CRISPR in HSC, including on target side effects, such as deletions, translocations etc. and recent work in Cell stem Cell, by the Dunbar lab on engraftment of HSC.

We thank the Reviewer for this comment. To comply with the Reviewer’s suggestion we have incorporated references no 15 entitled “Gene editing without ex vivo culture evades genotoxicity in human hematopoietic stem cells” and no 16 entitled: “Impact of CRISPR/HDR editing versus lentiviral transduction on long-term engraftment and clonal dynamics of HSPCs in rhesus macaques” and included the respective text on page 4, lines 142 and 143, on page 9, lines 374, 380 and 381.

  1. I am missing the work of the Hannover group on IVIM and recently SAGA assays as important safety assays for LV transduction in HSC

We thank the Reviewer for this comment. To comply with the Reviewer’s suggestion we have incorporated references no 54 entitled “Development of an in vitro genotoxicity assay to detect retroviral vector-induced lymphoid insertional mutants” and no 58 entitled: “Predicting genotoxicity of viral vectors for stem cell gene therapy using gene expression-based machine learning” and included the respective text on page 8, lines 348, and on page 9, lines 356 and 357.

  1. There are many more indications for which HSC based gene therapy is in clinical trails than authors discus, notably Artemis-SCID, RAG1-SCID and Fanconi anemia. This should be discussed and relevant work cited. Perhaps adding a table of ongoing trails for HSC GT would be useful

We thank the Reviewer for this comment. To comply with the Reviewer’s suggestion we have included Table 3 on page 10 and 11. We could not find an ongoing clinical trial for Fanconi anemia.

Reviewer 3 Report

Comments and Suggestions for Authors

Overall, the review is well written and comprehensively introduced the advancement in the gene delivery and transduction technologies for HSC gene therapy, with main advancements, remaining challenges, and future directions outlaid. It is ready to be accepted and published. But I believe the manuscript cab be further improved by addressing the below comments.

1. Could the authors please make a table or a diagram summarizing the clinical approved products for HSC gene therapy and their detailed technologies?

2. I am also expecting the most recent technologies such as nanoparticle and also CRISPR-mediated gene editing being introduced and discussed with more depth.

3. Line 299, should LMO2 gene be a tumor suppressor gene given its downregulated expression in lymphocytic leukemia?

4. Line 353, “HSCs transduction” should be “HSC transduction”.

Author Response

Overall, the review is well written and comprehensively introduced the advancement in the gene delivery and transduction technologies for HSC gene therapy, with main advancements, remaining challenges, and future directions outlaid. It is ready to be accepted and published. But I believe the manuscript cab be further improved by addressing the below comments.

We thank the Reviewer for this comment.

  1. Could the authors please make a table or a diagram summarizing the clinical approved products for HSC gene therapy and their detailed technologies?

We thank the Reviewer for this comment. As per the Reviewer’s suggestion, we have included Table 1 on pages 2 and 3.

  1. I am also expecting the most recent technologies such as nanoparticle and also CRISPR-mediated gene editing being introduced and discussed with more depth.

Addressing the Reviewer respectfully, we would like to mention that the current review focuses most on transduction enhancers and lentiviral mediated gene delivery to HSCs. It was not our aim to analyze in depth the current advancements for CRISPR and nanoparticles. However, to comply with the Reviewer’s comment, we have included the following references along with their mentions in the revised text:

- reference no 15 entitled “Gene editing without ex vivo culture evades genotoxicity in human hematopoietic stem cells” on page 4, lines 142 and on page 9, line 381;

-  reference no 16 entitled “Impact of CRISPR/HDR editing versus lentiviral transduction on long-term engraftment and clonal dynamics of HSPCs in rhesus macaques” on page 4, line 143 and on page 9, line 374 and 380;

- reference no 63 entitled “Nanoparticles targeting hematopoietic stem and progenitor cells: Multimodal carriers for the treatment of hematological diseases on page 11, line 437;

-  reference no 64 entitled “Lipid nanoparticles allow efficient and harmless ex vivo gene editing of human hematopoietic cells” on page 11, line 439.

  1. Line 299, should LMO2 gene be a tumor suppressor gene given its downregulated expression in lymphocytic leukemia?

We cordially thank the Reviewer for this “eagle eye” comment. We have corrected the sentence which can be now found on page 8, line 329.

  1. Line 353, “HSCs transduction” should be “HSC transduction”.

We cordially thank the Reviewer for this comment. We have deleted the paragraph that was previously located in line 353 and included a new paragraph with a relevant context on page 12, line 452.

Round 2

Reviewer 2 Report

Comments and Suggestions for Authors

The manuscript has improved enormously. The added figures and tables are useful. My only minor point left is to have the title reflect better  the focus on transduction enhancers

Author Response

Reviewer's comment:

The manuscript has improved enormously. The added figures and tables are useful. My only minor point left is to have the title reflect better  the focus on transduction enhancers

We thank the reviewer for this comment. We amended the title from: Advancements in hematopoietic stem cell transduction for gene therapy: a journey of progress

to 

Advancements in hematopoietic stem cell gene therapy: a journey of progress for viral transduction

These changes can be found in the title of the review in blue. 

We hope this amendment satisfies the Reviewer.